# Cancer Worry Distribution and Willingness to Undergo Colonoscopy at Three Levels of Hypothetical Cancer Risk—A Population-Based Survey in Sweden

**DOI:** 10.3390/cancers14040918

**Published:** 2022-02-12

**Authors:** Carolina Hawranek, Johan Maxon, Andreas Andersson, Bethany Van Guelpen, Senada Hajdarevic, Barbro Numan Hellquist, Anna Rosén

**Affiliations:** 1Department of Radiation Sciences, Oncology, Umeå University, SE-901 87 Umeå, Sweden; johan@maxon.se (J.M.); anden_16@hotmail.com (A.A.); bethany.vanguelpen@umu.se (B.V.G.); barbro.hellquist@onkologi.umu.se (B.N.H.); anna.rosen@umu.se (A.R.); 2Wallenberg Centre for Molecular Medicine, Umeå University, SE-901 87 Umeå, Sweden; 3Department of Nursing, Umeå University, SE-901 87 Umeå, Sweden; senada.hajdarevic@umu.se; 4Department of Public Health and Clinical Medicine, Family Medicine, Umeå University, SE-901 87 Umeå, Sweden

**Keywords:** cancer, oncology, cancer worry, cancer worry scale, colonoscopy, colorectal cancer, early detection of cancer, patient reported outcome measures

## Abstract

**Simple Summary:**

Cancer worry is a known health concern in cancer patients and people with a genetic predisposition to cancer. We measured how worried people, in general, are about developing cancer to describe levels in non-affected individuals. In total, 943 respondents completed a survey containing the Cancer Worry Scale (CWS) and hypothetical questions asking if they would attend a colonoscopy screening at a 5, 10, or 70 percent lifetime risk of developing bowel cancer. Unaffected individuals scored a mean of 9.46 on the six-item CWS. Women scored significantly higher than men (9.91 vs. 9.06). Women and parents had higher cancer worry than men and people without children when ruling out differences in education, age, and country of birth. People who worried more were also more inclined to undergo a colonoscopy screening, and intention increased with higher levels of hypothetical risk. These data may be helpful in future work on cancer worry and cancer prevention.

**Abstract:**

Purpose: We describe levels of cancer worry in the general population as measured with the Cancer Worry Scale (CWS) and investigate the association with colonoscopy screening intentions in three colorectal cancer risk scenarios. Methods: The data were sourced through a population-based survey. Respondents (*n* = 943) completed an eight-item CWS and questions on colonoscopy screening interest at three hypothetical risk levels. Results: Respondents without a personal cancer history (*n* = 853) scored 9.46 on the six-item CWS (mean, SD 2.72). Mean scores were significantly higher in women (9.91, SD 2.89) as compared to men (9.06, SD 2.49, *p* < 0.001). Linear regression showed higher cancer worry in women and those with children when controlling for education, age group, and country of birth. High cancer worry (six-item CWS mean >12) was identified in 25% of women and in 17% of men. Among those, 71% would attend a colonoscopy screening compared to 52% of those with low cancer worry (*p* < 0.001, 5% CRC-risk). Conclusions: The distribution of cancer worry in a general population sample showed higher mean scores in women, and levels overlapped with earlier findings in cancer-affected samples. Respondents with high cancer worry were more inclined to undergo a colonoscopy screening, and intention increased with higher levels of hypothetical risk.

## 1. Introduction

The fear of developing cancer, or the fear of cancer recurrence, is a known health concern in cancer-affected individuals, cancer survivors, and individuals with an increased hereditary risk of cancer [1,2,3,4,5]. Besides a negative impact on quality of life, mental health, and psychosocial wellbeing [2], cancer worry has been shown to modulate health-related behaviors such as lifestyle choices and screening participation [6,7,8]. Research suggests that cancer worry mainly facilitates, but in some cases also hinders, screening uptake [5,7,8,9,10]. Worry levels seem to correlate with a subjectively perceived risk of cancer [11], but the relationship between cancer worry and screening behavior seems to be complex and likely non-linear [7,12].

A commonly used instrument to assess cancer worry is the Cancer Worry Scale (CWS). The scale was originally developed as a four-item measure of frequency and severity of fear of developing cancer and has mostly been administered on high-risk individuals to assess how worry impacts mood and daily functioning [13,14,15]. The CWS was later expanded to include six and, finally, eight items [4,14], and has also been applied to assess fear of cancer recurrence [2]. The six-item CWS has been validated in cancer patients, and the suggested cut-off has been compared with the 42-item inventory on fear of cancer recurrence [4]. Although the six-item CWS show convergent and divergent validity and high internal consistency to detect fear of cancer, there is a lack of norm data from population-based samples. Moreover, to date, there is no standard definition of clinically significant cancer worry and no universally accepted approach to measuring this [3,4,5,14,16,17,18].

With the lack of normative population data and an ambiguous relationship between cancer worry and screening intentions, we addressed the following research questions:
(1)How is cancer worry distributed in the general population in Sweden?(2)Are cancer worry and the intention to undergo a colonoscopy screening associated?

## 2. Materials and Methods

We used a cross-sectional design administering an 8-item Cancer Worry Scale and three items on screening intentions in a population-based sample of the public in Sweden. Characteristics and subgroups were compared and tested for statistical significance. Results are presented separately for the 6-item CWS and the 8-item CWS and are separated for samples with and without individuals who had personal cancer experiences.

### 2.1. Data Collection

Survey data were sourced through an electronic questionnaire in an online research infrastructure administered by the Laboratory of Opinion Research (LORE) at Gothenburg University [19]. A number of subjects were selected to enable quantitative subgroup analyses. Subjects invited to this study had previously been randomly selected from the Swedish Population Register to join the Swedish Citizen Panel. The invitation included neutral text, non-revealing as to the topic of investigation, an estimate of how long the survey would take to complete, and how the collected data would be used and handled confidentially (Appendix A). The invitation was personally signed by the managing researcher and with the full name of the institution. Data were collected between the 12th of September and 7th of October 2018. Two electronic reminders were sent to non-responders on days 6 and 14 after initial survey distribution. Our questions were part of a larger set of questions with which we have previously reported on the public’s attitudes towards hereditary cancer risk communication [20].

### 2.2. Measurements

To measure cancer worry, respondents were asked to complete the 8-item four-point Likert scale (Appendix A). A total score ranged from 8 to 32, with a higher total score indicating greater worry about cancer and/or greater impact on mood and daily functioning. The adopted 8-item English Cancer Worry Scale [14] was translated into Swedish and then independently translated back into English to test retention of meaning and semantic coherence. Piloting of the questionnaire resulted in minor rephrasing. Due to the uncertain internal validity of questions 7 and 8 [9,21], results are presented with and without these last two items. The main text was based on results from the more widely used 6-item CWS in individuals without a personal cancer history, however, full 6- and 8-item results for all respondents are presented in the Appendix A. For an analysis using cut-offs between low and high cancer worry, the CWS scores of each dataset were grouped into low (≤11) and high worry (≥12) for the 6-item CWS and low (≤13) and high worry (≥14) for the 8-item CWS, according to previously defined cut-offs [4,9].

To investigate whether cancer worry affects cancer screening intention, we asked about respondents’ preferences to participate in a colonoscopy screening in three different hypothetical scenarios (Appendix A). Each scenario included a colorectal cancer risk level with a corresponding screening recommendation: an average population risk of 5% with one-time colonoscopy, a moderately increased lifetime risk of 10% with a colonoscopy every fifth year, or a greatly increased lifetime risk of 70% with colonoscopy every second year. Intention to undergo a colonoscopy was rated on a four-point Likert scale where “Yes, absolutely” and “Yes, I think so” were considered as positive responses and “No, I don’t think so” and “No, absolutely not” were considered as negative responses.

Self-reported information about gender, age, education level, country of birth, and parental status was acquired from the Citizen Panel database. In addition, one background question addressed personal cancer history: “Have you ever been diagnosed with cancer?” Response options included “Yes”, “No”, or “Prefer not to say”.

### 2.3. Statistical Methods

Categorical variables were described with counts and proportions and compared using chi-square tests. Continuous variables were described with medians, means, and standard deviations (SD) and were compared using *t*-tests and one-way ANOVA tests. To analyze factors associated with cancer worry, we applied a multivariable linear regression model with the 6-item CWS score as the dependent variable. Checking the model assumptions, we found that the distribution of the CWS score was skewed; therefore, the CWS score was log-transformed in the model. A *p*-value below 0.05 was considered statistically significant. The statistical software package R, version 3.5.2, was used for data analysis and creation of figures [22].

## 3. Results

Of the 1800 subjects invited, 943 respondents completed all eight cancer worry items (52%). Gender distribution did not differ between respondents and non-respondents, but older age, higher educational level, being born in Sweden, and having children were overrepresented among respondents (Appendix A). Among all respondents, 853 reported that they had no personal cancer history (90%). Moreover, 82 reported a previous cancer diagnosis and eight did not want to disclose a personal cancer history.

### 3.1. Distribution of Cancer Worry and Possible Determinants

#### 3.1.1. CWS Scores in Respondents without a Personal Cancer History

In respondents without a personal cancer history, the mean six-item CWS score was 9.46 (SD 2.72, Table 1). Both mean CWS scores and the proportion of respondents scoring in the higher interval were higher for women (9.91, SD 2.89 and 25%) compared to men (9.06, SD 2.49 and 17%). In the univariable analyses, CWS scores did not differ by age, educational level, having children, or country of birth (Table 1).

In a multivariable linear regression with log-transformed CWS scores as an outcome variable, female gender (9.3%, *p* < 0.001) and having children (4.9%, *p* = 0.04) were associated with increased cancer worry. Higher education (4.6%, *p* = 0.05) had a borderline significant association with lower cancer worry while age and country of birth did not (Figure 1).

In the eight-item CWS, results were similar to those from the six-item CWS, but total scores were higher due to two additional items (Appendix A).

#### 3.1.2. CWS Score in the Full Population-Based Sample

Mean CWS scores and the proportion of respondents scoring in the higher interval were significantly higher in women (10.10, SD 3.09 and 27%) as compared to men (9.17, SD 2.59 and 18%, *p* < 0.001 and *p* = 0.002, respectively). Respondents with a personal cancer history scored significantly higher on the six-item CWS than the larger unaffected group (mean 11.00 vs. 9.46, *p* < 0.001). CWS scores did not differ by age, educational level, having children, or country of birth (Appendix A). In the eight-item CWS, the results were similar to those from the six-item CWS, but total scores were higher due to two additional items (Appendix A).

### 3.2. Cancer Worry and Intention to Participate in Colorectal Cancer Screening

The distribution of cancer worry scores was skewed to the left (mean 9.08, median 8) among respondents with no or little intention to undergo a colonoscopy compared to those positive toward colonoscopy (mean 10.09, median 10, *t*-test, *p* <0.001) (Figure 2).

The proportion of respondents who intended to undergo a colonoscopy increased with the increasing level of hypothetical colorectal cancer risk (Chi2 test, *p* < 0.001). This trend remained when the data were stratified into high and low cancer worry subgroups (Figure 3).

## 4. Discussion

In the present study, we investigated cancer worry levels in a population-based sample of the Swedish public aged between 18 and 74 years. We outlined both six- and eight-item CWS data and provide sample characteristics on gender, age, education, having children or not, country of birth, and personal cancer history. We also reported the prevalence and factors associated with cancer worry in both individuals with and without a personal cancer history and in the full heterogenous population-based sample. Finally, we analyzed how cancer worry scores are associated with the intention to undergo a colonoscopy screening at three different levels of hypothetical lifetime risk of colorectal cancer. Each risk level included a description of the interval of surveillance recommended in the case of hereditary risk.

Comparison and interpretation of CWS-derived data from different contexts are challenging due to a large variation in instruments and number of CWS-items used throughout the literature [7,9]. Different phrasings of items, varying Likert scales [23,24], inconsistent presentation of the data [7], and lack of population-based norm data have hampered overall comparisons. To the best of our knowledge, this is the first study describing population-based norm CWS data in a Scandinavian population. This report of six- and eight-item CWS data with sample characteristics may help to inform future research on cancer worry, health-related behavior, and the clinical utility of the CWS scale.

### 4.1. Cancer Worry in the General Population and in Those without a Cancer History

The mean six-item CWS score in our full sample (9.61, SD 2.87) is similar to that reported in a somewhat comparable sample; this was a Spanish population of citizens aged 50 years and older recruited at primary health care and surveyed with the same six-item instrument (mean 9.3, SD 3.1) [25]. However, mean CWS scores were lower (7.1–8.8) in another study recruiting patients from primary health care for genetic testing for colorectal cancer risk in Australia [26].

In our data, respondents with a personal cancer history scored significantly higher on the six-item CWS than the larger unaffected group (mean 11.00). In previously published data on the six-item CWS, the mean scores among cancer survivors ranged from 9.0 for prostate cancer [4] to 8.4–9.2 for colorectal cancer [27] to 11.2 for breast cancer [4]. In this context, our subgroup of affected individuals appears to have scored in the upper interval of the cancer worry spectrum. Importantly, our subgroup of cancer-affected individuals was small (*n* = 82), self-reported, and not verified.

The mean eight-item CWS score in women was 13.96 (SD 4.14) and was 13.74 (SD 3.95) in women with no cancer history. In a recent study on curatively treated breast cancer patients, the mean values were 13.2–14.8 [28], whereas a study of women with newly diagnosed ovarian cancer reported a mean value of 17 [29]. In a Turkish study of women referred to a colonoscopy after a positive screening result, an even higher mean value of 20 was reported [30].

Hence, on the group level, there is an overlap in mean values of cancer worry between unaffected and affected populations as well as between different cancer types within affected populations. A recent review suggests that depression, anxiety, and distress among long-term cancer survivors do not significantly differ from the general population [31]. Although these concepts are not identical with cancer worry, they are closely related aspects in mental wellbeing and may hint at a pattern that is also valid for cancer worry.

### 4.2. Gender Differences in Cancer Worry

In previous research on cancer-affected populations [25,32] and in colorectal cancer screening participants [33], women report a higher cancer worry compared to men. This gender difference in cancer worry is confirmed in our population-based data. In a study using another instrument and sampling 2615 cancer survivors, female gender was found to be a strong predictor of fear of cancer recurrence while the type of cancer did not significantly affect fears [1]. Thus, gender appears to be an important determinant of cancer worry and should therefore be taken into consideration in future research on cancer worry, such as in the selection of matched controls, especially in patient groups with a skewed gender distribution.

### 4.3. What Constitutes “Clinically Relevant” Cancer Worry?

One previous approach to identifying “clinically relevant” cancer worry has been to use cut-offs dichotomizing respondents into high and low cancer worry groups [4,9]. In a re-validation study of the six-item CWS, severe fear of cancer recurrence was observed in 29.7% of patients with gastro-intestinal stromal tumors and colorectal, breast or prostate cancer survivors [4]. If the same suggested cut-off (≥12 total score) is applied to our six-item CWS data, 21% of the respondents without a personal cancer history would be categorized as having “high cancer worry” (Table 1).

Studies with the eight-item CWS and using the suggested cut-off (≥14 total score) [9] have identified high cancer worry in 31% of breast cancer survivors [9], 38% of CRC survivors [21], 33% of pancreatic ductal adenocarcinoma surveillance participants [16], and 47% of TP53-mutation carriers [34]. If applying the same suggested cut-off (≥14 total score) to the eight-item CWS in our study, 40% of the unaffected sample would be categorized as having “high cancer worry” (47% of women and 33% of men). The high proportion of individuals from the public scoring in the “high cancer worry” range raises the question of whether these cut-offs are suitable for determining “clinically relevant” cancer worry. This highlights the need for population norm data as a reference when interpreting CWS scores from patients [16,33] and warrants the further investigation of cancer worry levels in more heterogeneous samples.

### 4.4. Cancer Worry and Intention to Undergo Colonoscopy

Public awareness of CRC risk and screening benefits has been shown to be low in Europe, and, overall, Swedes report an average interest in screening compared to other European countries [35]. Cancer worry has been described both as a motivator and as a barrier for screening uptake [7,36]. Two explanations have been proposed for these discordant observations. Firstly, an inverted u-shaped relationship between cancer worry and screening uptake has been proposed where moderate worry levels correlate with higher screening uptake, while very low and severe cancer worry both reduce screening participation [7,25,37]. Secondly, different components of fear may act in opposite directions on screening adherence. Fear of the screening modality itself (i.e., colonoscopy exam) could reduce screening uptake, while general worry about cancer seems to motivate screening participation [8,17,38]. The relationship between worry and screening intentions is thus influenced by a number of parallel factors, several of which seem to be mediated through constructs in the health belief model [12].

In our study, high cancer worry was associated with higher intention to undergo a colonoscopy. As we do not see any un-linear association between cancer worry and colonoscopy intention (Figure 2), our data support the notion that the CWS instrument may in fact measure general cancer worry/fear of cancer recurrence without the confounder “fear of the screening modality”. In addition, intention to undergo screening increased with higher hypothetical lifetime risk of colorectal cancer despite a clear description of the corresponding increases in frequency of colonoscopies. This is in line with results from a clinical RCT study in Scotland where reported risk level perception was a significant factor in colonoscopy screening intention [39].

### 4.5. Study Limitations

This study has several limitations that should be considered. Of the 1800 individuals invited, 990 respondents entered the survey, 13 blank answers were excluded, and another 34 were incomplete and excluded due to missing data. The participation rate of 52% (*n* = 943) is comparable with current rates in other publicly funded surveys internationally [40] but nonresponse bias may still have affected our data. To minimize the risk of self-selection bias, the invitation to the survey was neutral and did not disclose the topic beforehand as the term “cancer” has a known deterring effect [17], perhaps especially so in people with higher cancer worry.

Generalizations obtained from our data should be made with caution as our sample characteristics differed compared to both non-respondents and the Swedish population at large (Appendix A). Our sample had an overrepresentation of people of older age, being born in Sweden, and with higher education.

The choice of age group in our sample and the screening modality in our scenarios was governed by the aim to offer norm data for comparison of cancer worry and screening interest in both public samples and for specific high-risk patient groups who may be offered colonoscopies due to increased cancer risk.

We measured intention to undergo a colonoscopy in hypothetical colorectal cancer risk scenarios, and consideration of the so-called “intention-behavior gap” described in social psychology is warranted [41]. As positive intentions are realized only half of the time while negative intentions are almost always predictive of behavior, our findings should be considered in light of this possibility.

We also acknowledge that our cross-sectional design limits our analysis because recent data suggest worry levels can fluctuate over time. In a one-year follow-up study of women with breast cancer, over 50% of respondents reported both high and low cancer worry when completing regular monthly assessments [28]. However, in other larger reviews, the fear of cancer recurrence was found to be stable over time [2], and the heterogeneous nature of our population-based sample may display a more even trend of worry than the data from cancer patients who have recently experienced disease and emotional strain.

### 4.6. Clinical Implications

This report offers clinicians and researchers a resource for age- and gender-matched comparisons of cancer worry data as a baseline reference from a heterogeneous general population. As our data show a substantial proportion of the public scoring in the “high cancer worry” range, further investigation of the cut-offs for clinically relevant cancer worry may be warranted in order to direct supportive resources to patients with the greatest needs. Moreover, as recent results suggest that cancer worry may fluctuate during the course of cancer treatment [28], future studies using longitudinal designs would be of interest to investigate the possible changes in worry levels over time in affected patients and high-risk individuals. We agree with the authors of the longitudinal breast cancer study who suggest a stepwise model wherein repeated measures could identify those in need of further interventions due to severe cancer worry at levels negatively affecting quality of life and daily functioning.

## 5. Conclusions

In this cross-sectional study of a Swedish population-based sample, female gender and personal cancer history were associated with higher cancer worry. However, cancer worry levels in our unaffected population were comparable to previous findings in cancer patients and survivors, suggesting that the psychological aspects of worry about cancer affect both patients and the public. Among those with high cancer worry, a larger proportion reported an intention to participate in a colonoscopy screening exam, and the willingness to undergo a screening increased in scenarios with higher risks of cancer. Our data support the idea that elevated cancer worry motivates health-related screening intention.

## Figures and Tables

**Figure 1 cancers-14-00918-f001:**
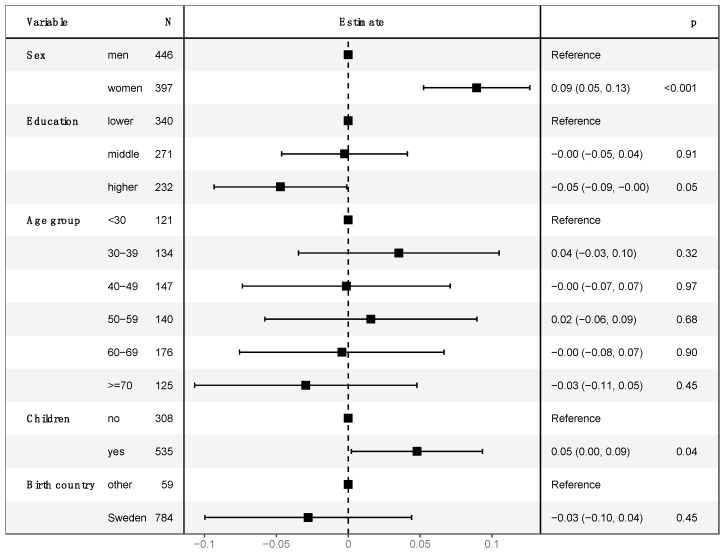
Multivariable linear regression of log-transformed 6-item CWS score in respondents without personal cancer history. The model is adjusted for sex (men/women), education level (lower, middle, higher), age group (<30 years/30–39 years/40–49 years/50–59 years/60–69 years/>69 years), children (yes/no), and country of birth (Sweden/other). The outcome of 6-item CWS score was log-transformed due to its skewed distribution. Intercept estimate was 2.172178 (*p* < 0.001).

**Figure 2 cancers-14-00918-f002:**
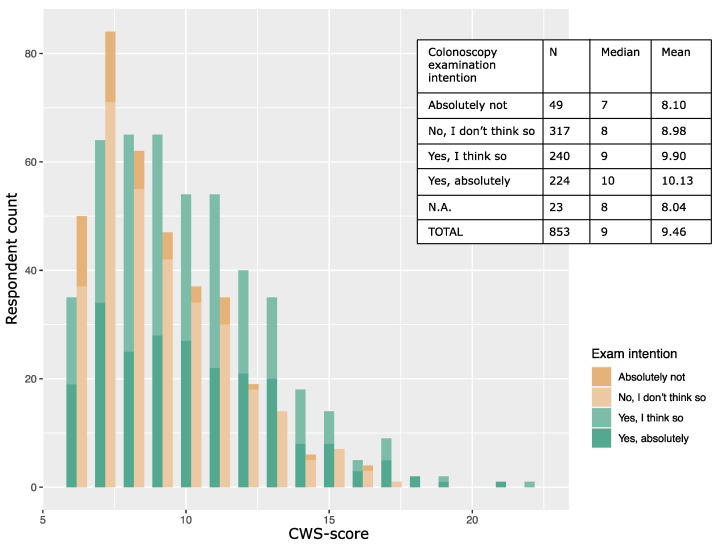
Distribution of cancer worry score grouped by intention to undergo a colonoscopy. Six-item CWS scores among respondents with intention to undergo a colonoscopy (green bars, *n* = 464, median 10, mean 10.01) and respondents without intention to undergo a colonoscopy (brown bars, *n* = 366, median 8, mean 8.86) in the scenario of having a 5% lifetime colorectal cancer risk.

**Figure 3 cancers-14-00918-f003:**
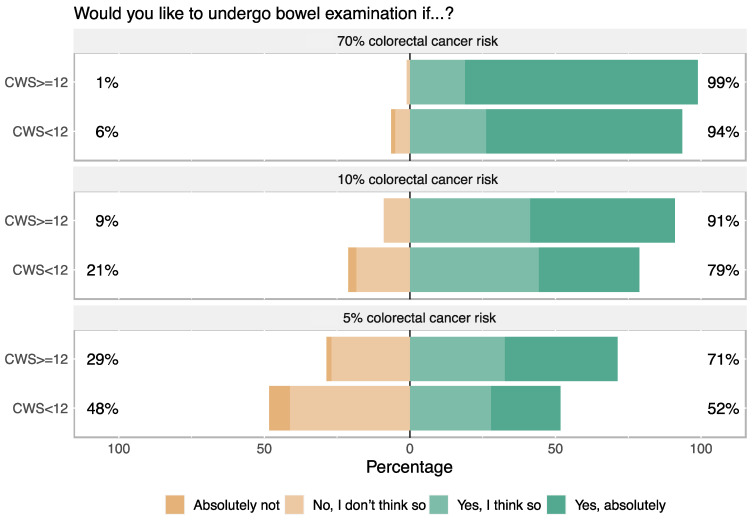
Intention to participate in colonoscopy screening at three levels of hypothetical colorectal cancer risk (*n* = 853, without cancer history). The scenarios presented three levels risk with the corresponding colonoscopy exam intervals: 5% lifetime risk with one-time colonoscopy, 10% lifetime risk with colonoscopy every five years, and 70% with colonoscopy every second year. Differences between the proportion of respondents with high and low cancer worry were significant in all three scenarios (*p* < 0.001 for bars in the two bottom panels and *p* = 0.025 for bars in the top panel).

**Table 1 cancers-14-00918-t001:** Six-item CWS score in participants without personal cancer history in a population-based sample (*n* = 853).

		Total	6-Item CWS Score			6-Item CWS Score Interval		
	Subgroup	N	Median (Min–Max)	Mean (Stand. Dev.)	*p*-Value *t*-Test/ANOVA	Low (CWS Score 6–11) N (%) ^†^	High (CWS Score 12–24) N (%) ^†^	*p*-Value Chi-Square Test
Total	-	853	9 (6–22)	9.46 (2.72)		674 (79)	179 (21)	
Gender	Women	403	9 (6–22)	9.91 (2.89)		301 (75)	102 (25)	
	Men	450	9 (6–19)	9.06 (2.49)		373 (83)	77 (17)	
					*p* < 0.001 ***			*p* = 0.004 **
Age	18–29	123	9 (6–19)	9.28 (2.78)		98 (80)	25 (20)	
	30–39	135	9 (6–22)	9.65 (3.18)		102 (76)	33 (24)	
	40–49	153	9 (6–17)	9.45 (2.76)		120 (78)	33 (22)	
	50–59	141	9 (6–17)	9.67 (2.62)		107 (76)	34 (24)	
	60–69	176	9 (6–17)	9.41 (2.47)		144 (82)	32 (18)	
	70–74	125	9 (6–18)	9.28 (2.53)		103 (82)	22 (18)	
					*p* = 0.74			*p* = 0.61
Education ^‡^	Lower	344	9 (6–22)	9.50 (2.69)		267 (78)	77 (22)	
	Middle	271	9 (6–21)	9.60 (2.81)		213 (79)	58 (21)	
	Higher	234	9 (6–17)	9.24 (2.67)		190 (81)	44 (19)	
					*p* = 0.31			*p* = 0.58
Country of birth ^§^	Sweden	785	9 (6–22)	9.43 (2.69)		622 (79)	163 (21)	
	Other	14	9 (6–19)	9.68 (3.08)		46 (77)	14 (23)	
					*p* = 0.55			*p* = 0.75
Children ^¶^	Yes	540	9 (6–21)	9.59 (2.65)		419 (78)	121 (22)	
	No	308	8 (6–22)	9.21 (2.80)		252 (82)	56 (18)	
					*p* = 0.06			*p* = 0.17

^†^ Numbers may not sum to 100 due to rounding. ^‡^ Education categorized into Lower (high school or less), Middle (up to 2 years at post-secondary level), or Higher (over 2 years at post-secondary level). NA (*n* = 4) not included. ^§^ Country of birth with response options: Sweden, Europe, or Outside Europe clustered into Sweden and Other. NA (*n* = 8) not included. ^¶^ Respondents’ answers to the question, “Do you have children?” NA (*n* = 5) not included. ** Indicate a *p*-value of ≤0.01 and *** indicate a *p*-value of ≤0.001.

## Data Availability

Both data and scripts used to generate the analyses in this study are available upon request. We encourage other researchers or institutes to use our population-based norm data where applicable.

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
