# Peer review of "Cancer Worry Distribution and Willingness to Undergo Colonoscopy at Three Levels of Hypothetical Cancer Risk—A Population-Based Survey in Sweden"

_cancers, 2022, doi:10.3390/cancers14040918_

Round 1

Reviewer 1 Report

This paper investigates the willingness of a population of Swedish patients to undergo colonoscopy at three levels of hypothetical cancer risk according to their self-reported Cancer Worry Scale. The authors show that there is higher cancer worry in women and parents and that these patients were more inclined to undergo screening colonoscopies and that their intention increased with higher levels of hypothetical risk. This article re-enforces the notion that certain segments of the population are more concerned about developing cancer than other and, therefore, are more receptive to cancer screening programs and their potential benefits. It is a large study with sound statistics. Its conclusions are most likely valid and generalizable and could be useful for further investigations into population-based strategies aimed at increasing screening rates in the future.

The authors do a nice job explaining the Cancer Worry Scale and how it has been validated. The article attempts to address the gap in knowledge about how cancer worry is distributed in the Swedish population.  There are a few methodological weaknesses that I think could be clarified. For one, out of 1800 subjects invited, only 943 respondents completed the survey. There is no discussion or data regarding why so many subjects did not complete the survey. It might be enlightening to poll a random sample of the subjects who did not complete the survey to see if their cancer worry was consistent with the subjects that did complete the questionnaire. For example, it might be that the non-respondents were excessively worried about cancer risk or less worried about cancer than the respondents and, if more subjects had responded, would alter the results considerably.  Also, there is very limited demographic data presented. I realize that the Swedish population is fairly homogenous with regards to race and socioeconomic status, but it would be interesting to present this data. In other more diverse countries, the attitudes towards cancer worry and inclination to undergo screening tests would be greatly affected by these variables.

In general, the article is accurately written and well referenced. The data is clearly presented. The authors appropriately acknowledge that population norms for cancer worry are not available, which makes it difficult to draw firm conclusions from this data.

Author Response

Answers to comments by Reviewer 1:

(also available in the attachment as PDF-file)

Yes

Can be improved

Must be improved

Not applicable

Does the introduction provide sufficient background and include all relevant references?

(x)

( )

( )

( )

Is the research design appropriate?

(x)

( )

( )

( )

Are the methods adequately described?

(x)

( )

( )

( )

Are the results clearly presented?

(x)

( )

( )

( )

Are the conclusions supported by the results?

( )

(x)

( )

( )

Comments and Suggestions for Authors

R1 comment 1: This paper investigates the willingness of a population of Swedish patients to undergo colonoscopy at three levels of hypothetical cancer risk according to their self-reported Cancer Worry Scale. The authors show that there is higher cancer worry in women and parents and that these patients were more inclined to undergo screening colonoscopies and that their intention increased with higher levels of hypothetical risk.

This article re-enforces the notion that certain segments of the population are more concerned about developing cancer than other and, therefore, are more receptive to cancer screening programs and their potential benefits. It is a large study with sound statistics. Its conclusions are most likely valid and generalizable and could be useful for further investigations into population-based strategies aimed at increasing screening rates in the future. 

Authors response: Thank you very much for reviewing our work and sharing your expertise to improve our manuscript. You present a very accurate summary, and we also hope this data can be of benefit to both clinical and research communities.

R1 comment 2: The authors do a nice job explaining the Cancer Worry Scale and how it has been validated. The article attempts to address the gap in knowledge about how cancer worry is distributed in the Swedish population. There are a few methodological weaknesses that I think could be clarified. For one, out of 1800 subjects invited, only 943 respondents completed the survey. There is no discussion or data regarding why so many subjects did not complete the survey.

Authors response: Thank you for highlighting this shortcoming. In the discussion paragraph “Methodological considerations” on lines 273-276 we comment briefly on the non-response rate of 48% and note that the topic of cancer might be a deterrent for some people to participate. However, since the invitation to our panel was neutral without revealing the topic, drop-out because of the “cancer-topic” at this stage is unlikely. In fact, out of 1800 subjects invited, 990 respondents participated (as in entered the survey). 13 were excluded due to missing data and another 34 were excluded due to incomplete responses/missing data, leaving 943 responses with all items completed (CWS and coloscopy interest). The remaining non-responders are most likely due to other reasons, since evidence suggest that non-response rates are not found to correlate with nonresponse bias; “Nonresponse rates “explain” only about 11 percent of the variation in different estimates of the nonresponse bias.” [1]. To elaborate on these issues, we have reviewed the discussion section to discuss the non-respondents and how they may influence survey results.

R1 comment 3: It might be enlightening to poll a random sample of the subjects who did not complete the survey to see if their cancer worry was consistent with the subjects that did complete the questionnaire. For example, it might be that the non-respondents were excessively worried about cancer risk or less worried about cancer than the respondents and, if more subjects had responded, would alter the results considerably. 

Authors response: The fact that non-responders would be more, or less, worried about cancer is a possibility, which we acknowledge. Indeed, a non-respondent poll would be very interesting to conduct. However, as this is a voluntary citizen panel survey, we do not have an option to re-contact non-responders and thus cannot produce such data.

We do present a comparison of characteristics of non-responders and respondents in Tables S5 and S6 in the supplementary material.

In addition, we have consulted our collaborators at Laboratory of Opinion Research at Gothenburg University for advice regarding the completion rate. A study from this year discusses the methodology of Citizen Panels for data collection and describes response rates to lie between 55-70% [2], making our 52% response rate not very far outside this range. The concern for rising non-response in surveys has also been highlighted in literature, but with the conclusion that non-response rates are not automatically correlated with non-response bias [1]. A recent methodological study concludes that “to date, there is little evidence that falling response rates have had large effect on the quality of survey estimates” and non-response does not create any noticeable bias until non-response rates reach over 70% (a RR of 30%) [3]. To nuance the discussion on potential influence on our results we have rephrased and elaborated on the possible reasons behind nonresponse in section 5.4 Study limitations.

R1 comment 4: Also, there is very limited demographic data presented. I realize that the Swedish population is fairly homogenous with regards to race and socioeconomic status, but it would be interesting to present this data. In other more diverse countries, the attitudes towards cancer worry and inclination to undergo screening tests would be greatly affected by these variables.

Authors response: Thank you for highlighting this fact. It is well known that factors like socioeconomic status have significant impact on health outcomes and behavior. Therefore, we have included education as an indicator for socioeconomic status in the demographics presented for our sample (See Tables S5 and S6, pages 7-8 in Supplemental information file). The Citizen Panel collect data on “Born in Sweden/ Other”, which we present, but unfortunately, we do not have explicit data on race. In Table 1, we present CWS-score in the different subgroups of educational level and country of birth.

R1 comment 5: In general, the article is accurately written and well referenced. The data is clearly presented. The authors appropriately acknowledge that population norms for cancer worry are not available, which makes it difficult to draw firm conclusions from this data.

Authors response: Thank you for your expertise evaluation. We hope the availability of this data can add to a growing body of evidence, thus enabling future conclusions to be drawn from a collective number of studies from different countries/cultures and populations.

Thank you again for the valuable feedback and constructive input on our work.

Sincerely

  1. Hawranek

________________________________________

  1. Groves, R.M.; Peytcheva, E. The Impact of Nonresponse Rates on Nonresponse Bias: A Meta-Analysis. Public Opinion Quarterly 2008, 72, 167-189, doi:10.1093/poq/nfn011.
  2. Siira, E.; Wolf, A. Are digital citizen panels an innovative, deliberative approach to cardiovascular research? Eur J Cardiovasc Nurs 2022, doi:10.1093/eurjcn/zvab132.
  3. Hedlin, D. Is there a 'safe area' where the nonresponse rate has only a modest effect on bias despite non‐ignorable nonresponse? International Statistical Review 2020, 88, 642-657, doi:10.1111/insr.12359.

Reviewer 2 Report

The major criticism is that screening is not offered to individuals less than 50 The survey should have been aimed at the 50 to 74 age group

The screening test offered is  faecal immunological test not colonoscopy willingnes to participate using this test should have been assesed

The response rate of  just over 50 per cent is dissappointing 

were the letters  of invitation signed and could it have been done better

The Table should have the N for each group

The data shows what we know already that gender age and level of education  predicts participation in screening problems but the data on cancer fear is novel

Author Response

Answers to comments from Reviewer 2:

(also available in the attachment as PDF-file)

Yes

Can be improved

Must be improved

Not applicable

Does the introduction provide sufficient background and include all relevant references?

( )

( )

(x)

( )

Is the research design appropriate?

( )

(x)

( )

( )

Are the methods adequately described?

(x)

( )

( )

( )

Are the results clearly presented?

( )

(x)

( )

( )

Are the conclusions supported by the results?

( )

( )

(x)

( )

Comments and Suggestions for Authors

R2 comment 1: The major criticism is that screening is not offered to individuals less than 50. The survey should have been aimed at the 50 to 74 age group.

Authors response: Thank you for highlighting this fact. We acknowledge this would be the most interesting design in the context of a solely population-based screening. There are two reasons we surveyed all age groups. First, we wanted to present normative data on cancer worry in the entire adult population and compare this with screening interest. This would not have been possible with a narrower age selection. Secondly, in the wider clinical perspective there are several (younger) patients who may be at increased risk of CRC due to several reasons (i.e., hereditary cancer syndrome) and for which there is limited data to refer to regarding intention and willingness to undergo health exams in a risk-reducing context.

In our perspective, it is thus vital to understand how cancer worry and attitudes are distributed in both younger and older populations.

R2 comment 2: The screening test offered is faecal immunological test not colonoscopy willingness to participate using this test should have been assessed.

Authors response: Thank you for this comment. Indeed, as you accurately point out, population screening often uses the FIT modality as the first line of screening test. As explained above, our aim in this study was not to look specifically at interest of population-based CRC-screening, but rather present a comprehensive overlook of how cancer worry is distributed across the population and in addition, exploring any correlation with colonoscopy interest, to make these results relevant as a reference population for cancer patients and high-risk individuals. We chose a scenario with this invasive type of exam in order to collect clear opinions for or against participation in this type of health decision.

We also wanted to provide colleagues and other research teams with normative population data, as different medical fields may want to compare results with a specific target group with increased cancer risk, due to e.g. hereditary factors connected with increased risk of CRC. To clarify these design choices, we have now added a paragraph under the Discussion section discussing these points more in detail.

R2 comment 3: The response rate of just over 50 per cent is disappointing.

Authors response: We agree and would have hoped for a higher completion rate of the full survey. However, when discussing the turnout with the methodological specialists at the Laboratory of Opinion Research at Gothenburg University (who manages the citizen panel) we have been informed that a response rate of 52% does not deviate from what can be expected in this type of public surveys, as it is a well-known fact that response rates have been dropping in the last years [3]. There is however no clear evidence that increasing non-response rates automatically translate to greater non-response bias. Non-response bias seem to depend mainly on other factors, and the overall evidence show that “to date, there is little evidence that falling response rates have had large effect on the quality of survey estimates” [3]. We elaborated the section 5.4 with details on these limitations and how they compare with international response rates.

R2 comment 4: Were the letters of invitation signed and could it have been done better.

Authors response: This is of course an important aspect in recruitment. There is always room for improvement and the format and content of the invitation may play a part in the respondents’ decision to participate in the survey. We have consulted our collaborating partner managing this data collection who is one of Sweden’s top academic research institutes in the field to retrieve a copy of the invitation used.

In our case, the wording of the invitation was neutral, non-revealing of the topic under investigation, explaining roughly how long the survey will take and how collected data cannot be traced back to an individual, but used in summarized form for research and scientific reports. The letter is signed personally with full name of the Associate Professor who is the Research Manager for the citizen panel and the full name of the public institution.

To add transparency to this part of the data collection, we have added a new appendix with a translated version of the letter of invitation to the survey (see attached Appendix A). 

In addition, we have also reviewed the relevant parts of the manuscript and elaborated on the type of invite in the “study limitations” section 5.4.

R2 comment 5: The Table should have the N for each group.

Authors response: Thank you for noting this. We have added an extra column with number of individuals for each subgroup in Table 1 now, so that it is consistent with the remainder of the data tables presented in the Supplementary material file.

R3 comment 6: The data shows what we know already that gender age and level of education predicts participation in screening problems but the data on cancer fear is novel.

Authors response: Thank you for highlighting the novelty in our data. Yes, we have seen similar respondent characteristics correlating with screening interest in other studies, but those data are often on specific patient groups or other selected homogeneous populations. As our aim was to produce norm-data for reference on a heterogeneous population-based sample the characteristics correlating to cancer worry and screening preferences were important to verify in our material as well. We have reviewed the sections of the discussion and conclusions referring to previous evidence on factors involved in screening interest and clarified that our data complements previous findings.

Thank you again for the feedback and input! 

Sincerely

  1. Hawranek

________________________________________

  1. Groves, R.M.; Peytcheva, E. The Impact of Nonresponse Rates on Nonresponse Bias: A Meta-Analysis. Public Opinion Quarterly 2008, 72, 167-189, doi:10.1093/poq/nfn011.
  2. Siira, E.; Wolf, A. Are digital citizen panels an innovative, deliberative approach to cardiovascular research? Eur J Cardiovasc Nurs 2022, doi:10.1093/eurjcn/zvab132.
  3. Hedlin, D. Is there a 'safe area' where the nonresponse rate has only a modest effect on bias despite non‐ignorable nonresponse? International Statistical Review 2020, 88, 642-657, doi:10.1111/insr.12359.

Round 2

Reviewer 2 Report

The authors have argued their case of including all sections of the population including the small numbers satisfactorily  Perhaps theyscreeningcould include a reference

Public awareness of risk factors and screening of colorectal cancer in Europe

Eurpean J Cancer Prevention 2004 13 257-264

Author Response

Review round 2:

Response to reviewer 2 comment

R2 comment 1: "The authors have argued their case of including all sections of the population including the small numbers satisfactorily  Perhaps theyscreeningcould include a reference - Public awareness of risk factors and screening of colorectal cancer in Europe Eurpean J Cancer Prevention 2004 13 257-264"

Answer: We have read the suggested article and find it an excellent reference and addition to our manuscript. We've added a sentence on risk awareness and screening interest on population level in the European context in this manuscript version 1.4 (second revision) under the section "4.4. Cancer worry and intention to undergo colonoscopy" on page 8. 

Thank you again for dedicating your time and expertise to improving our paper. Sincerely

/C. Hawranek on behalf of the full author team
